

# Impact of Coarse-Mode Aerosol on Jiangxi Warm Clouds Considering Different Updraft and Activation Intensities: An SBM-FAST Approach

Yi Li[1,2], Xiaoli Liu[1,2*]

[1] China Meteorological Administration Aerosol-Cloud and Precipitation Key Laboratory, Nanjing University of Information Science and Technology, Nanjing 210044, China.

[2] College of Atmospheric Physics, Nanjing University of Information Science and Technology, Nanjing 210044, China.

*Correspondence to*: Xiaoli Liu (liuxiaoli2004y@nuist.edu.cn)

**Abstract.**

The effects of different aerosol modes on warm clouds vary, with coarse-mode aerosols having a unique influence on cloud droplet growth and cloud-rain auto-conversion. Therefore, understanding the influence of coarse-mode aerosol concentrations on warm cloud formation and development is critical for improving weather prediction models and climate projections. This study uses the SBM-FAST scheme in the WRF model to assess how variations in coarse-mode aerosol concentrations (Ncm) affect the macro and micro characteristics of warm clouds in Jiangxi, China, focusing on the impacts under different updraft (W) and activation intensities through sensitivity experiments. Results indicate that higher Ncm enhances early-stage droplet number concentrations at the cloud base, accelerate cloud development. Increased Ncm also promotes earlier collision-coalescence processes, with more active formation and coalescence of larger droplets at higher Ncm concentrations. Yet, the response of cloud microphysics, like droplet concentration and relative dispersion ($\varepsilon$), to Ncm changes is not linear, depending on the combined effects of updraft strength and cloud droplet activation. Lower W/activation ratios lead to lower droplet activation in suboptimal supersaturation, reducing average size but enhancing $\varepsilon$ of cloud droplet. The number concentration of cloud droplet present initial decline then rise trend with increasing Ncm, which reflects the balance between the aerosol-

activation replenishment and collision-coalescence depletion of small size cloud droplet, illustrating the nonlinear influence jointly caused by aerosol activation and droplet interactions.

## 1 Introduction

Warm clouds and the accompanying cloud physical processes are critical in precipitation formation, cloud radiative properties, and climate. Therefore, a deep understanding and accurate simulation of warm clouds' formation, development, and microphysical processes are of great significance for climate research and numerical forecasting of precipitation (Rosenfeld et al., 2014; Zhao et al., 2017).

In previous studies on warm clouds, lots of studies have focused on the variation of cloud characteristics, such as cloud

droplet number concentration (Nc) and droplet size distribution. Grosvenor et al. (2018) pointed out a significant relationship between Nc, cloud optical thickness, and cloud top temperature. Wang & Lu (2022) and Xie et al. (2015) highlighted the significant impact of cloud droplet spectrum on cloud microphysical quantities such as effective radius and Nc of cloud droplet.

It is found that ε is an important parameter describing the width and distribution characteristics of the cloud droplet spectrum (Wang & Lu, 2022). On the one hand, ε affects the cloud's effective radius, altering the auto-conversion from cloud

to rain droplet, thereby causing variations in the precipitation process (Liu et al., 2005, 2006; Zhu et al., 2020; Lu & Xu, 2021; Wang et al., 2022; Wang et al., 2023; Yang et al., 2023). On the other hand, ε also links cloud-aerosol interactions, significantly impacting the climate system (Xie et al., 2017).

Given the critical role of ε in cloud microphysical processes, many scholars have explored the factors affecting its variation. Studies have shown that atmospheric humidity, turbulence intensity, and vertical aircraft intensity all influence ε

(Lu et al., 2013; Zhu et al., 2020; Kumar et al., 2017). Meanwhile, the sensitivity of ε to aerosol number concentration (Na) and its activation process has received particular attention. Researchers indicate that variations in Na have a complex and



profound impact on ε, extending to droplet nucleation, growth, and the ultimate precipitation process (Ma et al., 2010; Wang

et al., 2011, 2019). Through a comparison of aircraft observations and satellite data of warm clouds in both the Northern and

Southern Hemispheres, Liu et al. (2003) found that an increase in Na leads to a decrease in the effective radius of cloud droplets

and a narrowing of the cloud droplet spectrum, thereby reducing the ε. Kant et al. (2019) analysed aerosol observation data

collected in India from 2000 to 2017, found that strong updraft (W) containing large amounts of mineral dust aerosols could

activate more cloud droplets, increasing competition for water vapor and thereby narrowing the cloud droplet spectrum and

limiting the growth of large droplets. However, some studies have pointed out that an increase in Na might lead to an increase

in ε. Pandithurai et al. (2012) conducted in-situ aircraft measurements of cloud microphysical properties and aerosols over the

Indian subcontinent, showing that ε is positively correlated with Na. Anil et al. (2012) investigated the effect of aerosols on

cloud droplet number concentration and droplet effective radius from ground-based measurements over a high-altitude site

where clouds pass over the surface. The study also found a positive correlation between Na and ε. Moreover, it is shown that

increasing Na in clean tropical or marine areas broadens the cloud droplet spectrum, extends cloud lifetime, and enhances

precipitation (Liu et al., 2020).

Furthermore, the impact of Na on ε varies with aerosol particle size. Liu et al. (2022) used satellite data to find that an

increase in aerosol particles ranging from 0.1 to 2.5 micrometres, acting as cloud condensation nuclei (CCN), could suppress

precipitation and prolong the lifetime of marine warm clouds. Rosenfeld et al. (2002) noted that due to their larger size and

mass, increases in coarse-mode aerosol concentration (Ncm) could more effectively promote the coalescence process of cloud

droplets and increase precipitation intensity, thereby affecting the cloud droplet spectrum and precipitation efficiency.

Compared to fine-mode aerosols, Ncm may have a more significant and complex impact on cloud microphysical processes

and cloud dynamics, showing considerable variability in different studies (Ramanathan et al., 2001; Zhao et al., 2018). It is

shown that large aerosol particles with diameters exceeding 2μm, acting as giant CCN, can increase ε during the collision-

coalescence process, promoting the growth of cloud droplets (Yin et al., 2000; Jensen and Nugent, 2017). However, based on aircraft observations over the United Arab Emirates, Wehbe et al. (2020) found that no significant collision-coalescence

process presents in warm clouds despite the presence of giant CCN.

Although it has been discovered that variations in Na significantly impact ε of cloud droplets, with this impact varying according to different aerosol particle size mode, such relationships are significantly constrained by another crucial environmental factor—the intensity of W. As Kunnen et al. (2013) described, W specifically play a pivotal role in the formation and development of cloud droplets, with their intensity regulates the vertical movement of cloud droplets, thereby directly

affecting the coalescence process. Furthermore, the strengthen of W promotes the activation of CCN, increasing Nc and the size of cloud drops while simultaneously reducing the ε (Zhu et al., 2020).

Further research has revealed that the intensity of W is widely regarded as one of the primary drivers behind the complex responses of cloud microphysical properties to aerosol changes (Chen et al., 2016; 2018). As Lohmann et al. (2005) pointed out, under specific W conditions, the relationship between Nc and Na exhibits non-linearity pattern, influenced by

environmental states or conditions. At lower Na, Nc increases linearly with Na. However, as Na increases, the relationship between Nc and Na becomes sub-linear and tends towards stabilization.

Reutter et al. (2006) explored the intrinsic link between W and the non-linear Nc-Na relationship by numerical simulation experiments, categorizing the non-linear Nc-Na relationship into three intervals based on the ratio of W to Na: the aerosol-limited regime, the transitional regime, and the W-limited regime. In the aerosol-limited state, corresponding to a high W/Na

ratio, a higher degree of supersaturation, and a strong linear dependency between Nc and Na, with Nc shows a weak dependency on W. For the W-limited regime, where the W/Na ratio is low, exhibits reduced supersaturation and decreased correlation between Nc and Na but an increased correlation between Nc and W. The transitional regime lies between the other two, where Nc shows a sub-linear correlation between Na and W. Chen et al. (2016) further explored the W/Na relationship

with the effect of ε considered. It found that in the aerosol-limited regime, ε initially increases with Na. It peaks in the

transitional regime and then decreases as Na continues to increase in the W-limited regime. This pattern highlights the

significant regulatory influence of W on aerosol-cloud interactions.

In summary, under the background of climate change, variations in aerosols' physicochemical properties have

significantly impacted the microphysical properties of warm clouds, exhibiting notable differences across various regions and

cloud types. Among these, the mode of aerosols plays a particularly significant role in influencing the aerosol-cloud interaction,

with coarse-mode aerosols affecting cloud microphysics through complex mechanisms and displaying significant variability

across different studies. It is obviously that the intensity of W acts as another critical factor constraining the non-linear

interaction between clouds and aerosols, leading to complex responses of cloud microphysics to changes in Na.

Consequently, this study uses the SBM-FAST bin microphysics scheme within the Weather Research and Forecasting

(WRF) model to simulate a stratiform warm cloud event in Jiangxi, China. Through a series of numerical experiments, this

study examines the effects of increasing Ncm by 5, 50, and 500 times on the macro and microphysical properties of warm

clouds in the Jiangxi region, with consideration of the influences of W intensity on aerosol-cloud interactions. The results may

provide a deeper understanding and background support for the droplet spectrum characteristics and the impact regulation of

Ncm on warm clouds in Eastern China. The structure of the following text is as follows: the second section introduces the

numerical simulation setup, the data used for validating the simulation results, and the computation formulas involved in the

analysis; the third section validates the simulation results of the control experiment and reveals the impact of Ncm on cloud

macroscopic and microscopic physical quantities, as well as the response of cloud microphysical processes to changes in Na

under different W strengths and relative intensities of cloud droplet activation. The last two sections mainly include the

discussion and conclusions.

## 2 Model Introduction and Experiments Design

### 2.1 Simulation Setup

This study simulated a warm cloud event in Jiangxi, China on December 25, 2014, by use of the version 4.2 of Weather Research and Forecasting (WRF) model. Numerical experiments include one control experiment and three sensitivity experiments, with concentration of coarse mode aerosols modified. Apart from coarse mode aerosol concentration, the control and sensitivity experiments maintain consistent initial fields and simulation settings.

The simulation used the fifth generation of the European Centre for Medium-Range Weather Forecasts (ECMWF) global atmospheric reanalysis (ERA5) hourly pressure level data as the initial field, with a resolution of 0.25°×0.25°. A double-nested approach was adopted, with grid resolutions of 3km and 1km, respectively, and the innermost grid number being 376×376. The microphysics scheme employed was the SBM-fast bin scheme (FSBM-2); the boundary layer scheme employed the Mellor-Yamada-Janjic (Eta) Turbulence Kinetic Energy (TKE) scheme, the surface layer scheme adopted the Monin-Obukhov (Janjic Eta) scheme, and the land surface model used the unified Noah land-surface model. The shortwave and longwave radiation schemes were the (old) Goddard shortwave radiation scheme and the Rapid Radiative Transfer Model (RRTM) scheme, respectively.

The simulation area and nesting setup are shown in Figure 1, with the simulation duration from 18:00 on December 24, to 06:00 on December 25, 2014 (UTC). To reduce the impact of the WRF model spin-up, we excluded the first six hours simulation results to analysis. No precipitation was observed at the ground surface during the simulation period. The simulation area, located in Ganzhou City in the southern part of Jiangxi Province, lies in the upper reaches of the Gan River, between the southeastern coastal and central inland transition zones, under the control of the subtropical monsoon climate. The terrain is predominantly mountainous, with hills and basins.



The warm stratiform cloud process simulated in this study in Jiangxi Province is the same as our previous research,

which is part of a series of studies on the impact of varying aerosol concentrations on stratiform clouds in this region (Li et

al., 2024). The weather pattern for this process is identical to that in our previous research, and detailed weather analysis can

be found in the earlier paper (Li et al., 2024). This study focuses primarily on analysing the impact of different coarse-mode

aerosol concentrations on the microphysical properties of stratiform warm clouds.

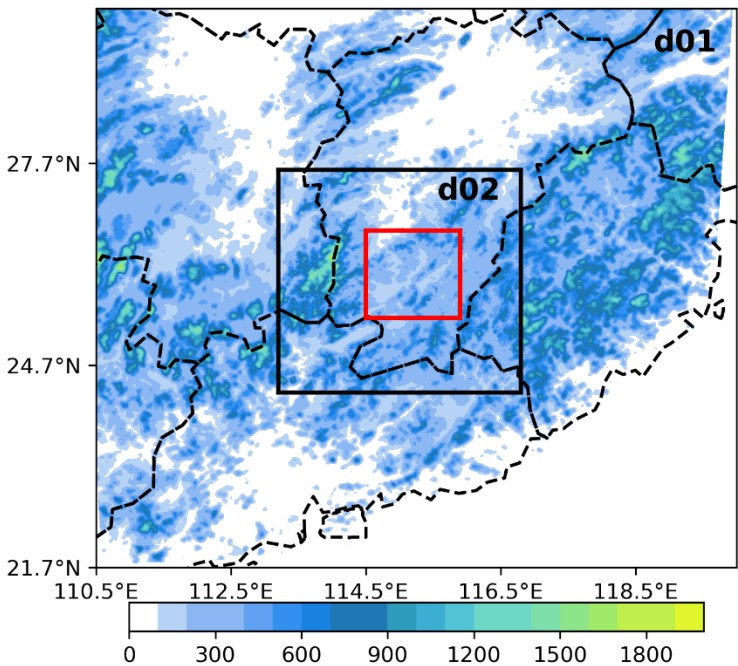

**Figure 1 The simulation area. The shaded portion in the figure represents the elevation of the terrain (in meters), and the area within the red frame is the scope of analysis in the following sections.**

**2.2 Introduction to Microphysics Scheme**

The FSBM-2 microphysics scheme used in this study is an evolution of the FSBM-1 scheme developed by Khain and

Lynn (2009). It represents simplifying and optimizing the original SBM-full scheme from the Hebrew University Cloud Model

(HUCM) (Khain and Sednev, 1996; Khain et al., 2000). Enhanced by Shpund et al. (2019), FSBM-2 has been demonstrated

by Han et al. (2019) to deliver more accurate simulation results.



The FSBM-2 scheme describes cloud and rain droplets through a unified size distribution consisting of 33 bins while two

different type of background aerosol (marine and continental) can be selected to act as CCN in different study area. In this

study, except for certain marine areas within the d01 domain, continental aerosol spectrum is adopted for the rest of the d01

area and the entire d02 region. There are 43 mass bins for aerosol spectrum distribution,   with the maximum radius of dry

aerosols is set to 2 micrometres. FSBM-2 scheme assumes the smallest aerosol size to be 0.003 micrometres and characterizes

the initial aerosol distribution through three log-normal distributions, corresponding to the nucleation mode (centered at 0.008

micrometres), accumulation mode (centered at 0.034 micrometres), and coarse mode (centered at 0.46 micrometres), using

supersaturation to calculate the nucleation process in the cloud.

**2.3 Setup of Sensitivity Experiments**

This work includes three sensitivity experiments and one control experiment (ORG). Initial aerosol concentrations set in

the control experiment, is shown in Table 1. For sensitivity experiments, with the coarse-mode aerosol concentration modified

to 5 times, 50 times, and 500 times its original value, respectively (Table 2). At the beginning of the simulation, the initial

aerosol spectrum for the analysis area is presented in Figure 2.

**Table 1 Initial aerosol spectrum settings for the control experiment.**

| Aerosol Type | Number Concentration (cm$^{-3}$) | Average Particle Size (μm) |
|---|---|---|
| Nucleation Mode | 1000.000 | 0.008 |
| Accumulation Mode | 800.000 | 0.034 |
| Coarse Mode | 0.720 | 0.460 |

**Table 2 Initial aerosol spectrum settings for the sensitivity experiments.**

| Experiment | Nucleation Mode (cm$^{-3}$) | Accumulation Mode (cm$^{-3}$) | Coarse Mode (cm$^{-3}$) |
|---|---|---|---|



| Experiment 1 (CM5) | 1000.000 | 800.000 | 3.600 |
| Experiment 2 (CM50) | 1000.000 | 800.000 | 36.000 |
| Experiment3 (CM500) | 1000.000 | 800.000 | 360.000 |

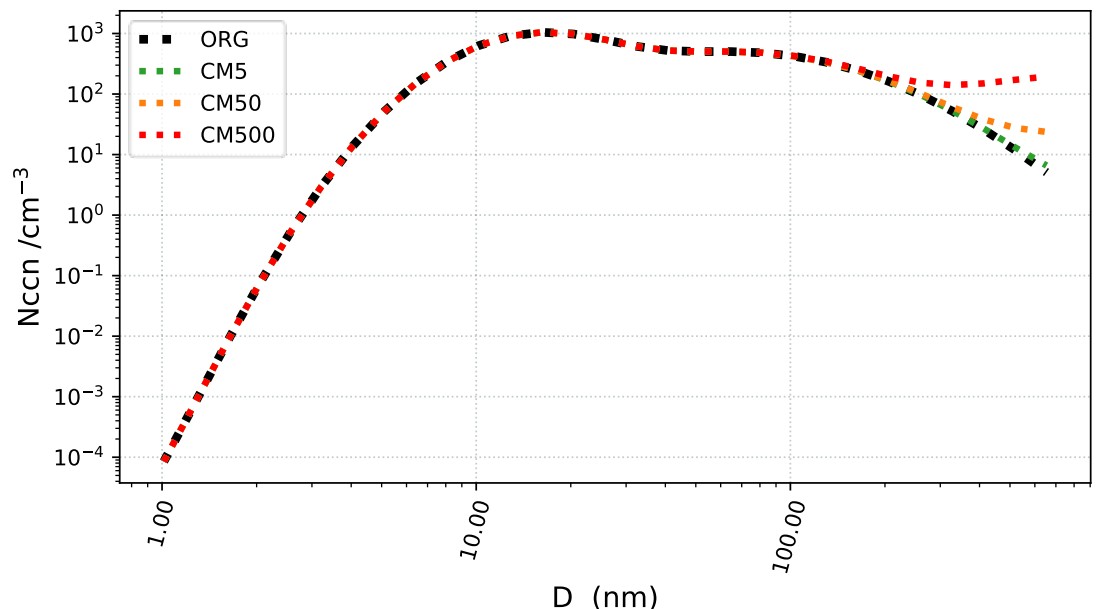

**Figure 2 Initial aerosol spectrums for the control and sensitivity experiments.**

## 2.4 Introduction of Observation Data

### 2.4.1 Introduction of Aircraft Observation Data

The aircraft observation data used in this study originate from a flight observation mission conducted over Jiangxi, China, on December 25, 2014. The observation flight area was located above Ganzhou City in Jiangxi Province, with coordinates ranging from 114.0°E to 117.0°E and 25°N to 27°N. Figure 4 displays the flight path of this aircraft observation. The aircraft

took off from Ganzhou Airport and conducted an observation flight around Ganzhou City, adopting a mode of initial ascent,

plane flight, and final spiralling descent flight path for observations.

### 2.4.2 Introduction to Satellite-Observed Cloud Top Temperature Data

This study utilized cloud top temperature data in standard format from the FY-2F meteorological satellite provided by the

National Satellite Meteorological Center (NSMC). The FY-2F satellite's scanning radiometer includes five channels. This

article's cloud top temperature data are derived from the VISSR-II channel onboard the FY-2F satellite. The spatial resolution

of this data is 5 km, with a temporal resolution of 1 hour, and the effective data range is from 0 to 400 K.

### 2.5 Cloud Droplet Spectrum Parameters Calculation

This study analysed the evolutions of cloud droplet spectrum characteristics, including mean cloud droplet radius (Rm),

the volume-weighted radius of cloud droplets (Rv), the auto-conversion threshold function value (T), the cloud droplet

spectrum relative dispersion (ε), and the cloud droplet activation intensity (Fbs). Detailed formulas for these calculations are

provided in Supplement.

## 3 Results Analysis

### 3.1 Simulation Results Validation

To validate the simulation results of the control experiment, we selected simulation results at 02:30 and 03:30 UTC, while

cloud development was most vigorous, to compare the simulated cloud top temperatures with those observed by the FY-2F

satellite during the same period. As shown in Figure 3, satellite observations indicate that warm clouds were primarily

concentrated in central and southern regions of Jiangxi, with cloud-top temperatures ranging from 0 to 15°C. At 02:30, the

warm clouds within the flight observation area were distributed in an east-west band. By 03:30, the warm clouds in the flight

observation area showed a dispersed trend, with cloud top temperatures ranging from 0 to 10°C. Compared to satellite

observations, the distribution and variation trend of simulated clouds top temperature generally matched the observation results,

although the simulated cloud top temperatures were slightly higher. Indeed, differences in cloud distribution between the

observations and simulations largely stem from resolution differences (satellite data have a horizontal resolution of 5 km × 5

km, whereas the control experiment has a resolution of 1 km × 1 km), where higher resolution for numerical simulation aids

in capturing more detailed cloud distribution features.

The aircraft observations on December 25, 2014, provided detailed information on microphysical properties within the

cloud. To validate the simulation result, a comprehensive cloud penetration process from 04:10 to 04:20 was selected (Figure

4). The Clw, Nc and Rm from the simulation during the same period were compared with the aircraft observations at the same

altitudes. To minimize the impact of differences in vertical resolution between the model and aircraft observations, cloud

height normalization was performed, setting the cloud base height to 0 and the cloud top height to 1.

As shown in Figure 5, the Clw in the control experiment initially increases and then decreases with altitude, which

aligns well with the aircraft observations. Although there are some differences in the vertical trend of Nc, the control

simulation generally maintains consistency with the aircraft observations regarding the magnitude of number concentration.

Additionally, in both the control simulation and aircraft observation data, Rm increases with altitude, display similar vertical

distribution characteristics.

Despite the similar vertical distribution trends of Clw and Nc in the simulation results and the aircraft observations,

notable differences still exist in the magnitudes of the data. These discrepancies mainly arise because aircraft observations

capture instantaneous data along specific flight paths through the clouds.  The numerical simulation results represent the

average values of cloud microphysical quantities at the same latitude, longitude, and altitude as the aircraft's cloud

penetration process from 04:10 to 04:20.  Thus, although the vertical distribution trends are similar, specific values of Clw



and Rm show variations due to the regional coverage of cloud grid points in numerical simulations compared to direct

aircraft measurements.

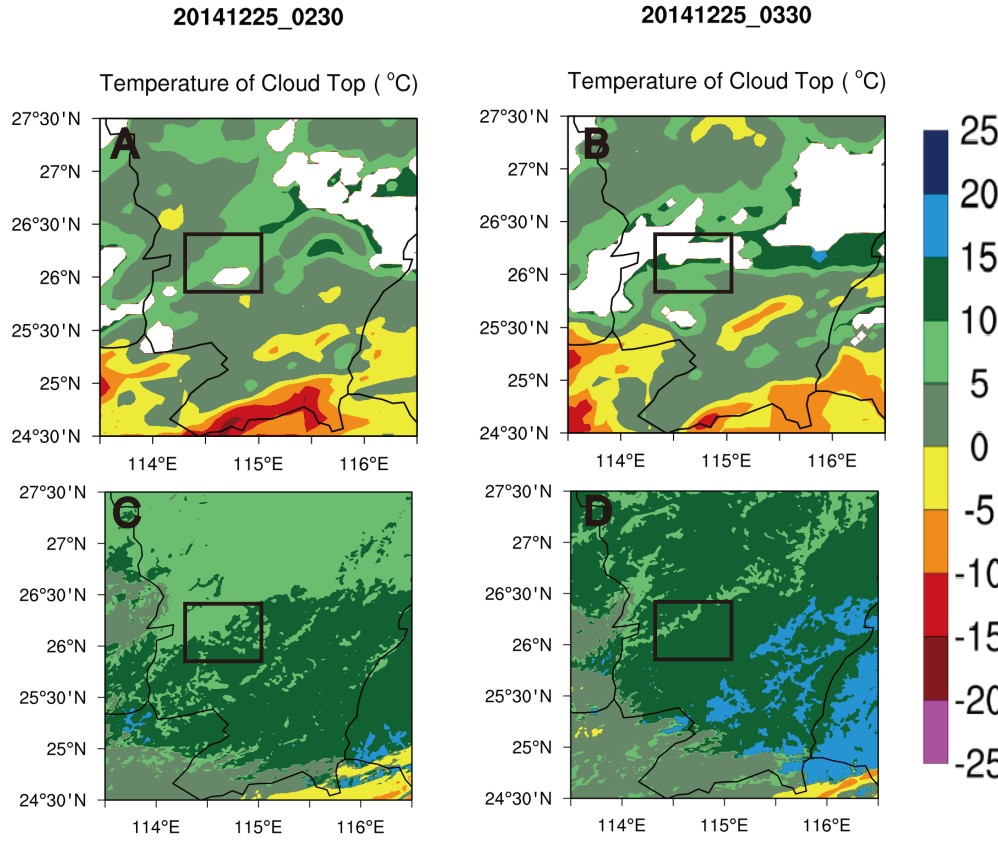

**Figure 3 FY-2F satellite observed (A at 02:30, B at 03:30) and control experiment simulated (C at 02:30, D at 03:30) cloud-top temperatures on December 25, 2014 (unit: °C). The black box indicates the aircraft observation area.**



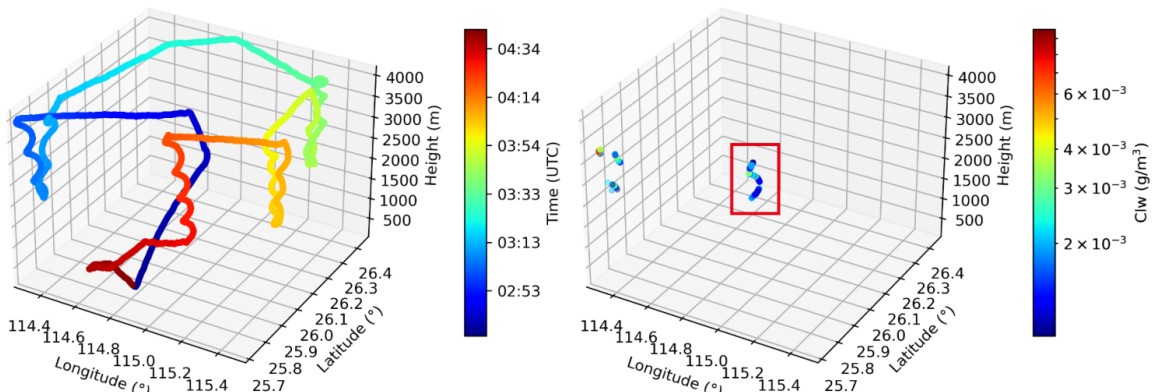

**Figure 4 Aircraft flight trajectory and cloud liquid water content (Clw) within the cloud region along the observation path. The red box indicates a comprehensive cloud penetration process from 04:10 to 04:20 UTC.**

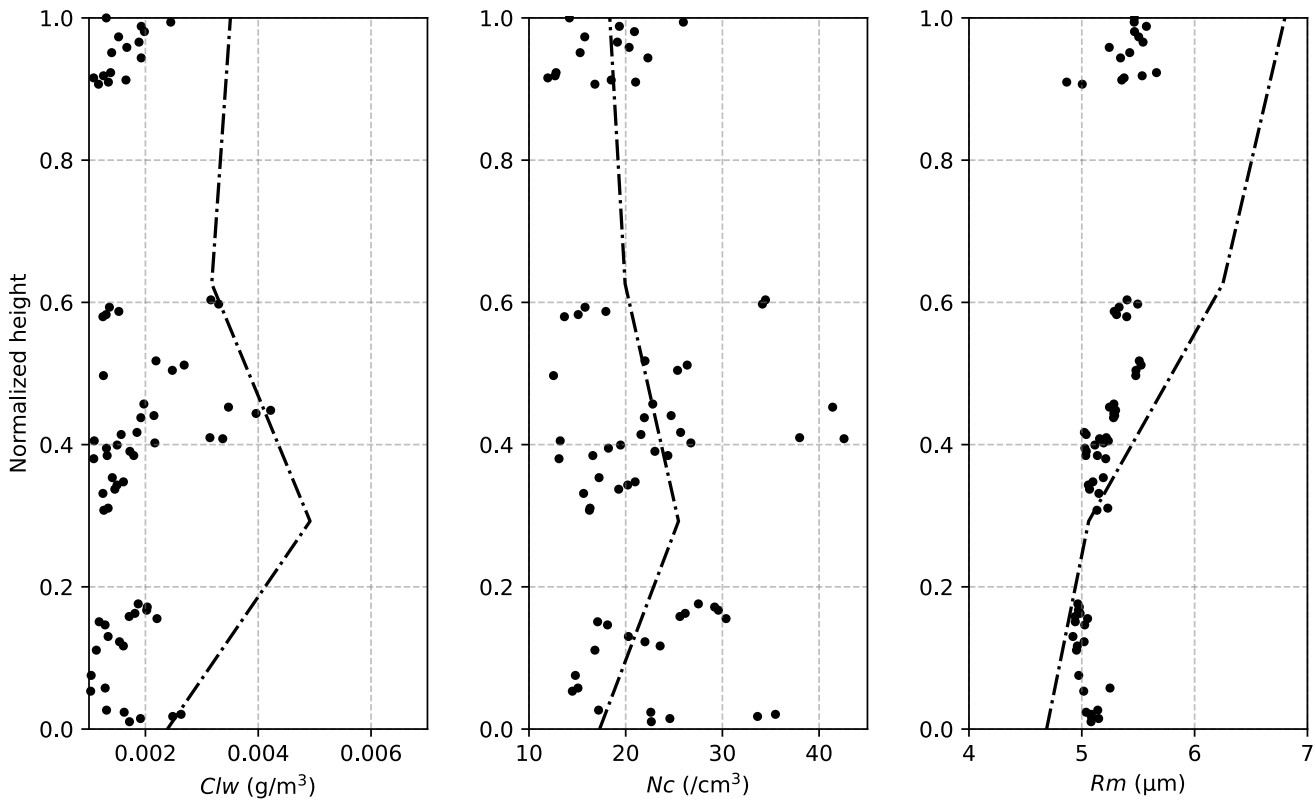


**Figure 5 Aircraft observations (scatter points) of cloud liquid water content (Clw, in g/m³), cloud droplet number concentration (Nc, in cm⁻³), and cloud droplet mean radius (Rm, in µm) on December 25, 2014 at 04:10 to 04:20 UTC, compared with model simulations (black dashed lines) of Clw, Nc, and Rm.**

## 3.2 Vertical Distribution of Cloud Microphysical Quantities

Figures 6-8 illustrate Nc, Rm, and Clw variations with time and altitude. Based on evolutions in Nc and Clw, along with the timing of peak values, we divided this warm cloud process into an early development stage (00-03 hours) and a vigorous development stage (03-05 hours). After 05:00 UTC, cold clouds began to appear in the study area, marking the transition of the warm cloud process towards a mixed-phase process.

In the early development stage, Nc, Clw and Rm all show increasing trends over time. For the control experiment, during the early development stage, Nc gradually increasing and extending upwards over time(Figure 6). During the vigorous development stage, Nc significantly increases in the cloud base region, indicating an enhanced cloud droplet activation process at this stage. At the same time, Rm increases with height (Figure 7), and Clw gradually increases in the mid and upper cloud layers (Figure 8).

For the sensitivity experiments, the impact of Ncm on cloud microphysical properties shows significant differences at different heights and time stages. During the early development stage, in the CM5 experiment, Rm and Clw slightly increase, but the impact on Nc is minimal. In contrast, in the CM50 and CM500 experiments, Nc significantly increases in the cloud base region, especially in the CM500 experiment, where the increase in Nc is most pronounced, accompanied by a decrease in Rm in the cloud base region, indicating that more small cloud droplets are activated, which is consistent with the findings of Ramanathan et al. (2001) and Yang et al. (2023).

During the vigorous development stage, Nc and Clw significantly increased in the cloud top region in the CM50 and CM500 experiments, particularly in the CM500 experiment, where both Nc and Clw are substantially higher than in the control experiment. In addition, the increased Ncm concentration also leads to an earlier development of cloud tops above 3.5 km.



This finding aligns with van den Heever et al. (2006) results, which noted that larger CCN particles begin to enhance cloud

235     liquid water content in the later stages of cloud development.

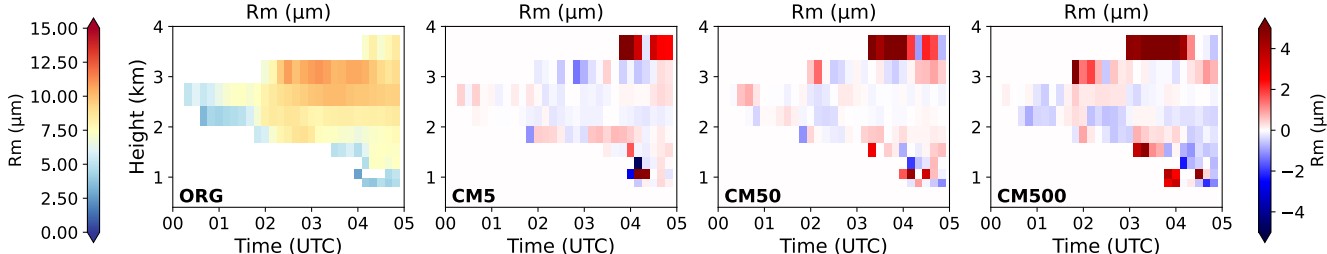

**Figure 6 The variations of cloud droplet number concentration (Nc, cm⁻³) with time (UTC) and altitude (km) within the study area**

**of different experiments. From left to right, the panels represent the control experiment (ORG) and differences between the three**

**sensitivity experiments (CM5, CM50, CM500) and the control experiment. The left side color shading indicates the magnitude of**

240     **Nc in the control experiment. The right-side color shading indicates the difference of Nc between the sensitivity experiments and**

**the control experiment, with red showing values higher than the control experiment and blue showing values lower than the**

**control experiment.**

**Figure 7 The variations of averaged cloud droplet radius (Rm, µm) with time (UTC) and altitude (km) within the study area of**

245     **different experiments. From left to right, the panels represent the control experiment (ORG) and differences between the three**

**sensitivity experiments (CM5, CM50, CM500) and the control experiment. The left side color shading indicates the magnitude of**

**Rm in the control experiment. The right-side color shading indicates the difference of Rm between the sensitivity experiments and**

**the control experiment, with red showing values higher than the control experiment and blue showing values lower than the**

**control experiment.**





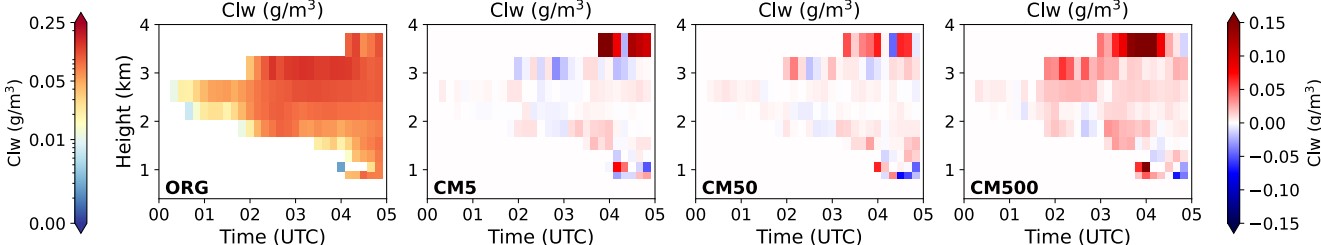

**Figure 8 The variations of averaged cloud liquid water content (Clw, g/m³) with time (UTC) and altitude (km) within the study area of different experiments. From left to right, the panels represent the control experiment (ORG) and differences between the three sensitivity experiments (CM5, CM50, CM500) and the control experiment. The left side color shading indicates the magnitude of Clw in the control experiment. The right-side color shading indicates the difference of Clw between the sensitivity experiments and the control experiment, with red showing values higher than the control experiment and blue showing values lower than the control experiment.**

### 3.3 Characteristics of Cloud Droplet Size Distribution

Given the significant differences in cloud microphysical quantities along the vertical height, this study normalized the distribution of heights within the cloud layer and thereby divided the cloud layer into three regions: cloud top region (C_high), cloud middle region (C_mid), and cloud base region (C_low). The differences in cloud droplet spectrum within these specific areas are further analysed (as shown in Figure 9). The cloud droplet spectrum in the C_low and C_high regions exhibited more apparent sensitivity to the variation of Ncm. Specifically, in the C_low region, the number of cloud droplets with diameters in the range of 16 - 44 μm increased in the CM50 experiment. In the CM500 experiment, the number of small cloud droplets with diameters less than 12 μm and large cloud droplets with diameters greater than 32 μm increased. However, when the Ncm concentration was only increased by 5 times, the cloud droplet spectrum in the cloud base region showed little difference compared to the control experiment. Whereas in the cloud top region, under increased Ncm concentration conditions, the cloud droplet spectrum showed different variations across the sensitivity experiments. When Ncm increased by 5 and 500 times, the number of large droplets (28 - 52 μm) was significantly increased. When Ncm increased by 50 times, there was no significant change in the cloud droplet spectrum in the cloud top region.



270   Figure 10 presents the probability distribution of cloud droplet number concentration concerning droplet diameter. As the

cloud system develops, the cloud droplet spectrum in the control experiment gradually broadens, with the peak concentration

of droplets within the 9-24 μm diameter range increasing, resulting in a unimodal distribution. When Ncm increases by five

times during the early development stage, the number concentration of droplets within the 9-15 μm diameter range increases,

but the spectrum width does not change significantly. In contrast, in the CM50 and CM500 experiments, the impact of

275 increased Ncm on the droplet spectrum during the early development stage is significant. At 01:00 UTC, the droplet spectrum

transitions from a unimodal distribution in the control experiment to a bimodal distribution, with a notable broadening of the

spectrum. The number concentration of droplets within the 4-15 μm diameter range increases, especially in the CM500

experiment, where the number concentration of small droplets with diameters less than 6 μm significantly increases during the

early development stage. During the vigorous development stage, the concentration of medium-sized droplets decreases in the

280 CM50 experiment, while the concentration of droplets within the 6-24 μm diameter range increases in the CM500 experiment.

   Consistent with the results of this study, Chuang et al. (2009) also pointed out that in the early development stage of

clouds, as the Ncm concentration increases, the number of activated small droplets increases, causing the peak value of the

cloud droplet spectrum to shift towards smaller sizes. The average droplet size correspondingly decreases, which is particularly

pronounced in the CM500 experiment. However, in the vigorous development stage, as the Ncm concentration increases, not

285 only does the number concentration of small droplets increase, but also the proportion of large droplets within the 15-24 μm

diameter range increases.



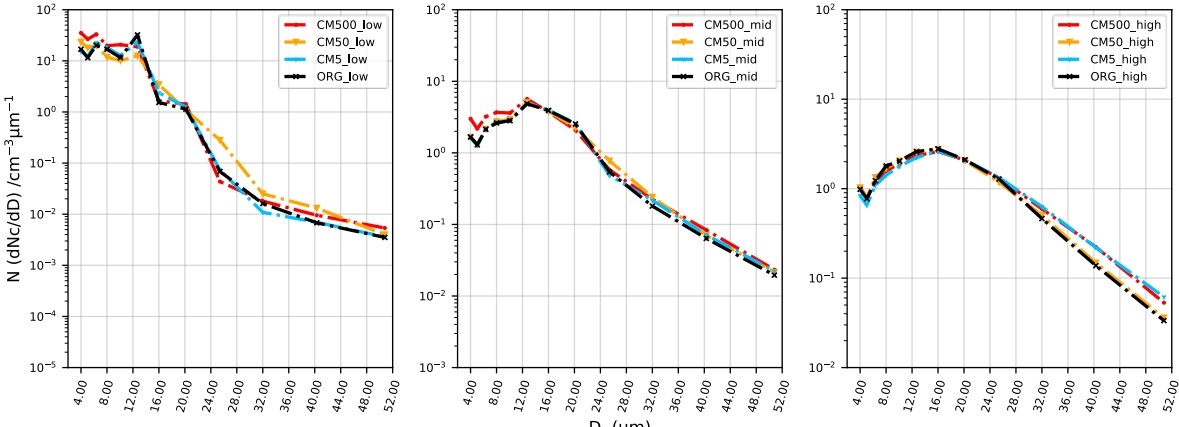

**Figure 9 Cloud droplet spectrum at different part of cloud within the study area. Here, 'low' represents the cloud base area, 'mid' indicates the mid-section of the cloud, and 'high' denotes the cloud top region.**

290

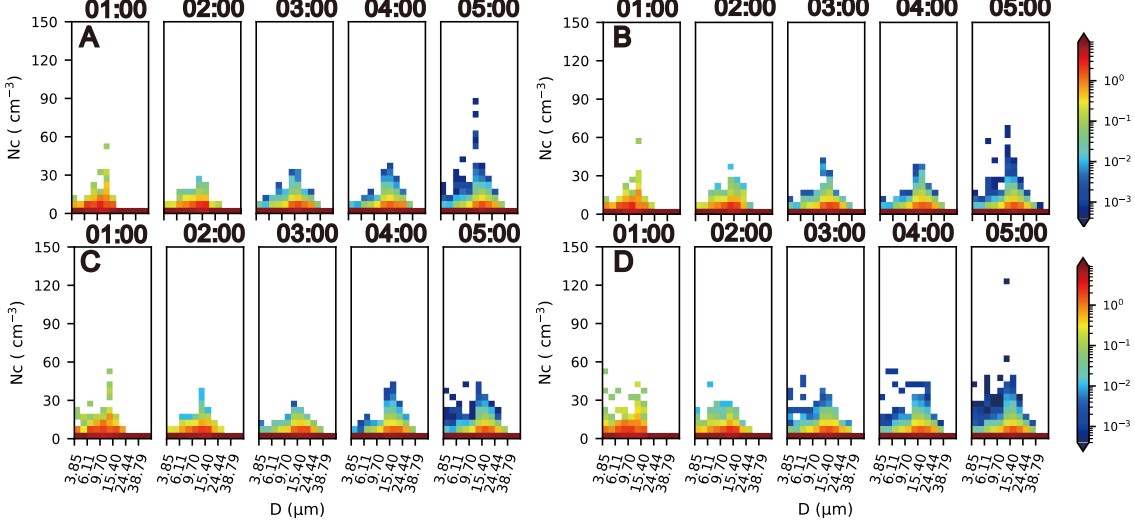

**Figure 10 Distribution of cloud droplet number concentration PDFs for different experiments. The panels represent the control experiment (ORG) and three sensitivity experiments (CM5, CM50, CM500). The color shading indicates the magnitude of probability.**

### 3.4 Microphysical Characteristics of Cloud Droplet Spectrum

### 3.4.1 Vertical Distribution of Microphysical Characteristics of Cloud Droplet Spectrum

As shown in Figure 11, the vertical distribution of ε exhibits complex temporal variations. In the early development stage, ε is relatively low, with high values concentrated at the cloud base. As the cloud system develops into the vigorous development stage, ε significantly increases above the 3 km altitude range, with peak values appearing in the top region.

As Ncm concentration increases, the peak value of ε at the cloud top also increases. However, at the cloud base, a significant enhancement in ε is only presented when Ncm concentration increases to 500 times that of the control experiment. Similar to the trend of ε, T values are relatively low during the early development stage and increase as the cloud system develops (Figure 12). The peak T values appear in the cloud top region during the vigorous development stage, corresponding to the most intense stage of cloud development. At the cloud base, the high T values align with those of ε, and the distribution of Rv (Figure 13) at the cloud base also matches the variations of T values. However, the relationship between ε, T values, and Rv is not monotonic at the cloud top. In addition, for the CM500 experiment, compared to the control experiment, strong collision-coalescence processes with T values greater than 0.5 occur earlier.

The study by Tas et al. (2012) indicated that dlnClw/dt can be used to identify the rate of change in the transformation of water vapor into cloud droplets within a cloud. A dlnClw/dt greater than zero typically signifies that condensation growth and/or activation processes dominate the change of cloud droplet liquid water content. In contrast, a dlnClw/dt less than zero suggests a reduction in Clw, which could be related to evaporation processes. According, the relative strengths of condensation growth and cloud droplet activation at different stages of cloud development is analysed by calculating the changes in dlnClw/dt across three sensitivity experiments.

As shown in Figure 14, during the early stages of cloud development, the increase in Clw is primarily contributed by cloud droplet activation and condensation growth processes. In the CM50 and CM500 experiments, the cloud droplet activation



at the cloud base increases, with dlnClw/dt being higher compared to the control experiment, accompanied by a decrease in Rv.

In the vigorous development stage of the cloud, in the middle to upper layers of the cloud, the growth in cloud droplet size is more dependent on the collision-coalescence process. At the same time, Rv significantly increases with the enhancement of collision-coalescence intensity. In the cloud base region, with increasing aerosol concentrations (CM5, CM50, CM500), dlnClw/dt increases, especially in the CM500 experiment. This indicates that more cloud droplets are activated in this area in the CM500 experiment. The increase in concentrations of small droplets leads to a greater difference in droplet size, resulting in increased dispersion (Figure 11). At this altitude, the ε shows a positive correlation with T values, as the increase in droplet size differences promotes the occurrence of cloud droplet collision-coalescence processes. In the middle to lower layers of the cloud, even though collision-coalescence activities are vigorous, the extensive activation of smaller droplets leads to an increase in the relative dispersion of cloud droplet size distribution, which is reflected in the increase in ε.

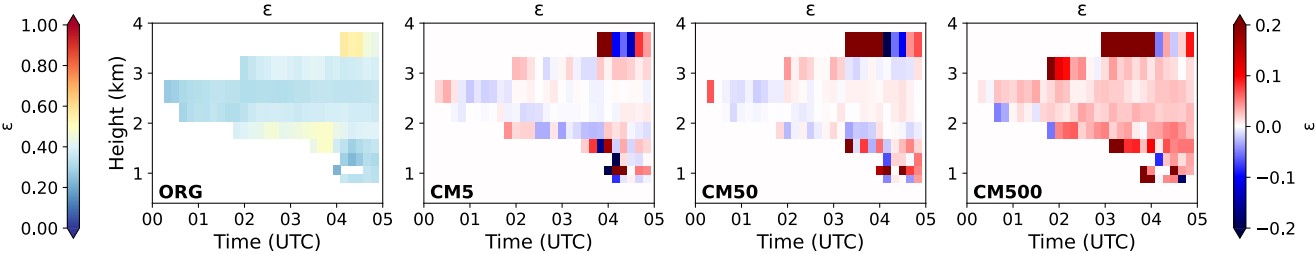

**Figure 11 Distribution of cloud droplet spectrum relative dispersion (ε) over time (UTC) and altitude (km). From left to right, the panels represent the control experiment (ORG) and differences between the three sensitivity experiments (CM5, CM50, CM500) and the control experiment. The left side color shading indicates the magnitude of ε in the control experiment. The right-side color shading indicates the difference of ε between the sensitivity experiments and the control experiment, with red showing values higher than the control experiment and blue showing values lower than the control experiment.**



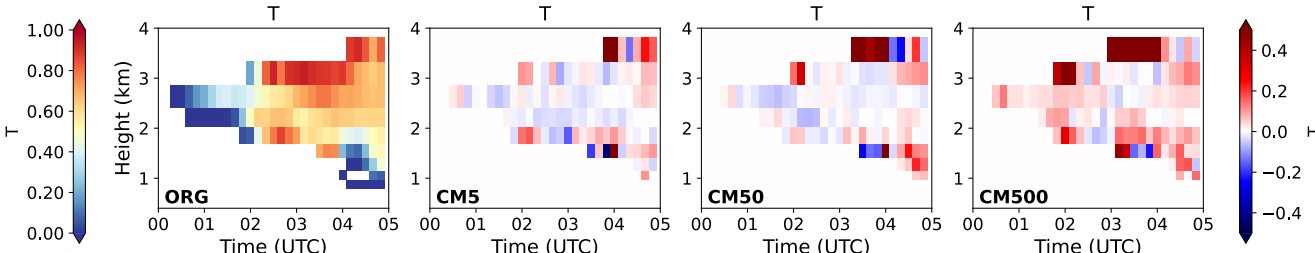

**Figure 12 Distribution of cloud droplet collision-coalescence intensity (T) over time (UTC) and altitude (km). From left to right, the panels represent the control experiment (ORG) and differences between the three sensitivity experiments (CM5, CM50, CM500) and the control experiment. The left side color shading indicates the magnitude of T in the control experiment. The right-side color shading indicates the difference of T between the sensitivity experiments and the control experiment, with red showing values higher than the control experiment and blue showing values lower than the control experiment.**

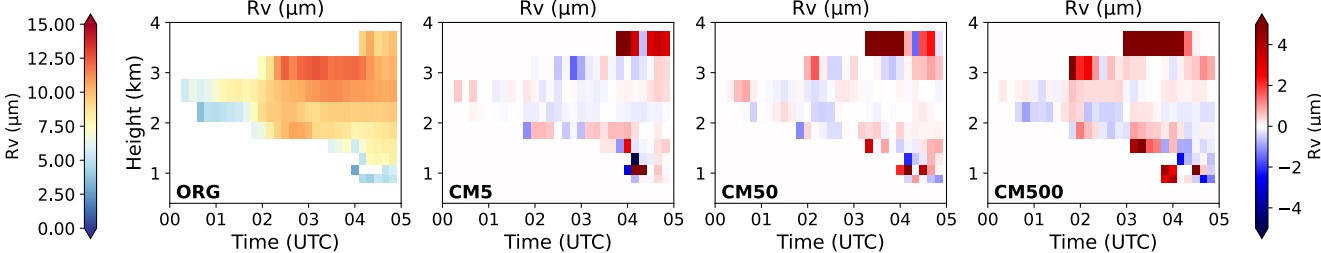

**Figure 13 Distribution of cloud droplet volume-weighted mean diameter (Rv, μm) over time (UTC) and altitude (km). From left to right, the panels represent the control experiment (ORG) and differences between the three sensitivity experiments (CM5, CM50, CM500) and the control experiment. The left side color shading indicates the magnitude of Rv in the control experiment. The right-side color shading indicates the difference of Rv between the sensitivity experiments and the control experiment, with red showing values higher than the control experiment and blue showing values lower than the control experiment.**



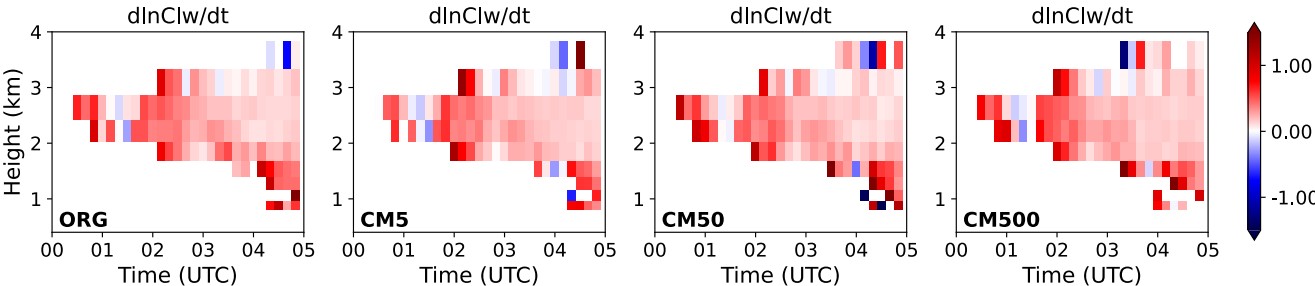


**Figure 14 Distribution of the logarithmic rate of change of liquid water content (dlnClw/dt) over time (UTC) and altitude (km). From left to right, the panels represent the control experiment (ORG) and three sensitivity experiments (CM5, CM50, CM500) as differences from the control experiment. The color shading indicates the magnitude of dlnClw/dt.**

### 3.4.2 Analysis of the Relationship between Cloud Droplet Spectrum Characteristics

Lu et al. (2006) and Reutter et al. (2009) highlighted that the significant impact of aerosol concentration on cloud microphysical properties is not only related to aerosol concentration but also controlled by supersaturation and closely associated with the microphysical processes. Chen et al. (2016, 2018) demonstrated that the relative strength of W and cloud droplet activation significantly affects cloud microphysical processes, resulting in noticeable differences in ε of cloud droplet spectrum. Therefore, this section analyses the changes in the relationship between cloud droplet spectrum characteristics under

different Wand cloud droplet activation strengths (Fbs) (Lu et al., 2020), dividing them into two intervals based on the relative strength of W and Fbs (Lu et al., 2020), as shown in Figures 15-18. The area above the black dashed line represents the "High Ratio Zone" (HRZ) where W/Fbs > 1, while the area below it represents the "Low Ratio Zone" (LRZ) where W/Fbs < 1.

It can be found that, as Lu et al.(2020) pointed out, cloud droplet spectrum characteristics, such as ε, are significantly influenced by W and Fbs, with apparent differences exist across different regions. For instance, larger droplets mainly

concentrate in the HRZ within the area where W ranges from 0 to 0.6 m/s (Figure 15). Under a fixed W, an increase in Fbs leads to a decrease in droplet size but an increase in relative dispersion. Varble et al. (2023) indicated that the W determines the maximum supersaturation that cloud droplets can achieve, to some extent representing the level of supersaturation.



Therefore, in the HRZ with strong W, high supersaturation conditions favor the condensational growth of cloud droplets. The difference in condensation growth rates between small and large droplets leads to a homogenization of droplet sizes, reducing

ε (Figure 17). In the control experiment, isolated maximum values of Nc exist in the region where W > 0.8. This is due to the high supersaturation and high W conditions in this area, leading to excessively active cloud droplet activation. A large number of small droplets compete for water vapor, limiting droplet size growth and resulting in a smaller Rv. The more uniform droplet sizes lead to a reduction in ε and lower collision-coalescence intensity. This phenomenon aligns with findings in stratiform cloud studies by Korolev et al. (2016) and Chen et al. (2016). Additionally, the formation of large droplets significantly

promotes the coalescence between cloud droplets, and the rapid consumption of small droplets is one of the main reasons for the inverse relationship between ε and Rv.

Similar to the HRZ, within the LRZ, the proportion of small droplets increases as the Fbs increases, leading to a decrease in Rv. However, different from the HRZ, ε - T is directly proportional within this zone, both increasing with a decrease in Rv. This phenomenon indicates that under low supersaturation conditions, although the differences in condensation growth rates between large and small cloud droplets tend to equalize droplet sizes, the enhanced activation of small droplets still leads to

an increase in ε.

Moreover, as shown in Figure 18, Nc is strongly constrained by W/supersaturation, with peak concentrations primarily occurring when the W exceeds 0.5 m/s and increasing with stronger W in the HRZ. supersaturation limits the peak cloud In contrast, in the LRZ, lower supersaturation limits the peak cloud droplet concentration, with peak values far lower than in the

HRZ and decreasing with Fbs.

With an increase in Ncm concentration in the HRZ, the required W to achieve the same Nc shows a significant decreasing trend. This indicates that under higher coarse mode aerosol concentrations, the level of supersaturation needed to maintain the same number of cloud droplets is lower. This is consistent with findings by McFiggans et al. (2006).





However, Nc exhibits a nonlinear response to changes in Ncm concentration. With an increase in Ncm concentration, the

maximum droplet number concentration first decreases, then increases (Figure 18). Several complex microphysical processes

may drive this non-monotonic behaviour. When the Ncm increases by 5 times, a rise in aerosol concentration is insufficient to

affect cloud droplet activation. At the 50 times increase in Ncm, the activation of more small droplets enhances ε (Figure 17).

This higher collision-coalescence intensity results in the rapid consumption of small droplets (Figure 16), reducing the

maximum Nc. When the Ncm is further increased to 500 times, even though collision-coalescence intensity is significantly

elevated, the involvement of more aerosol particles in activation allows for rapid depletion yet replenishment of small droplets

within the cloud. Consequently, compared to the control experiment, the Nc significantly increases, as does the ε.

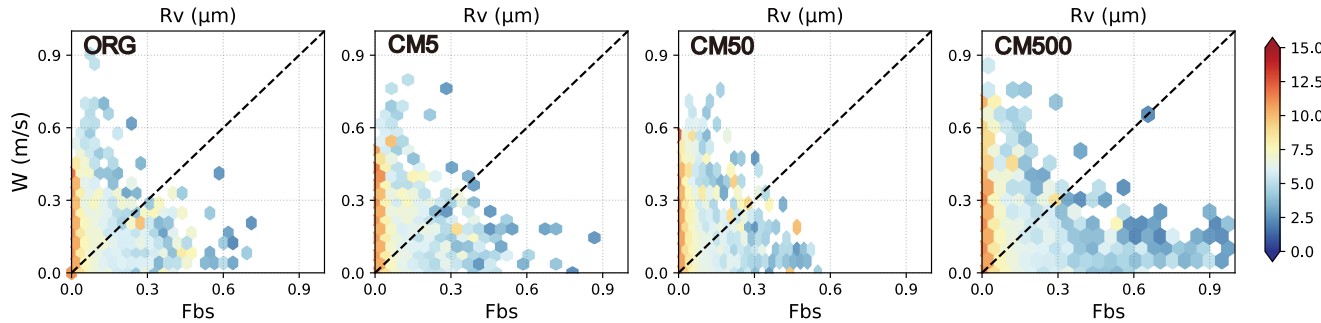

**Figure 15 Distribution of cloud droplet volume-weighted mean diameter (Rv, μm) as a function of updraft strength (W, in m/s) and activation intensity (Fbs). Panels ORG, CM5, CM50 and CM500 represent simulation results under different coarse-mode aerosol concentrations, respectively. The color of the dots indicates the magnitude of Rv value. Dashed lines represent the relationship line where W/Fbs=1, indicating equal ratios of updraft strength to activation intensity.**




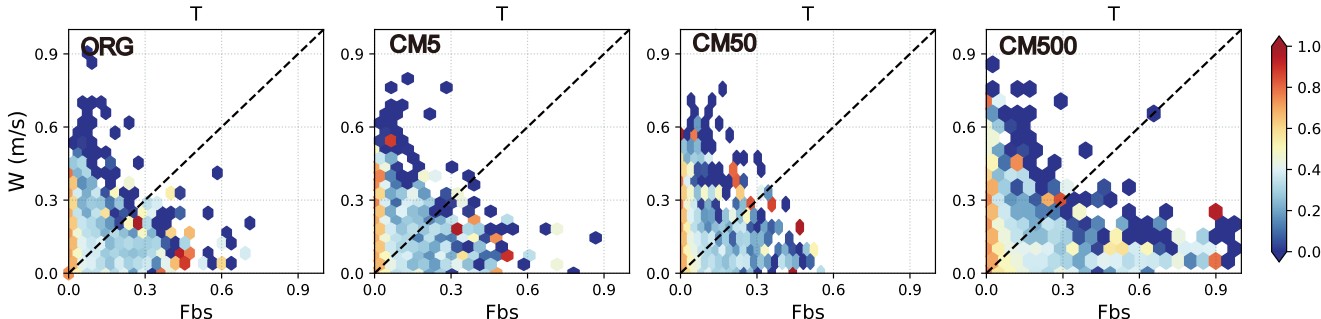

**Figure 16 Distribution of collision-coalescence intensity (T) as a function of updraft strength (W, in m/s) and activation intensity (Fbs). Panels ORG, CM5, CM50 and CM500 represent simulation results under different coarse-mode aerosol concentrations, respectively. The color of the dots indicates the magnitude of collision intensity. Dashed lines represent the relationship line where W/Fbs=1, indicating equal ratios of updraft strength to activation intensity.**

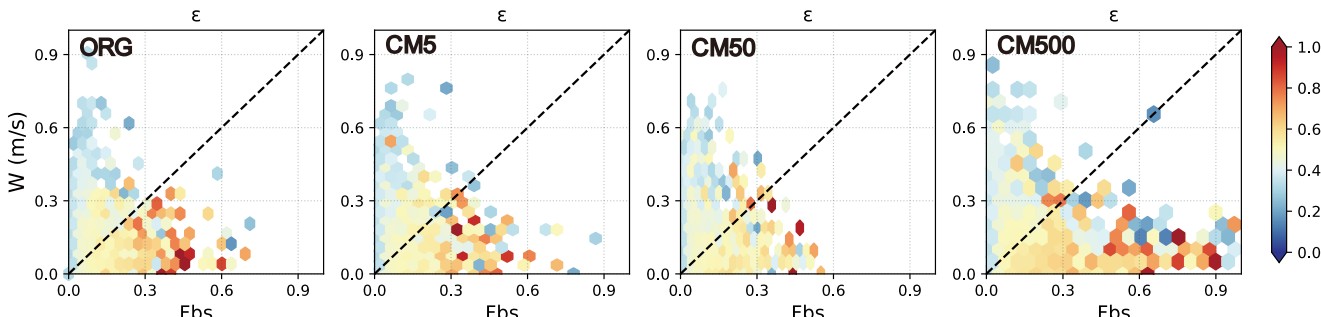

**Figure 17 Distribution of cloud droplet spectrum relieve dispersion (ε) as a function of updraft strength (W, in m/s) and activation intensity (Fbs). Panels ORG, CM5, CM50 and CM500 represent simulation results under different coarse-mode aerosol concentrations, respectively. The color of the dots indicates the magnitude of ε value. Dashed lines represent the relationship line where W/Fbs=1, indicating equal ratios of updraft strength to activation intensity.**

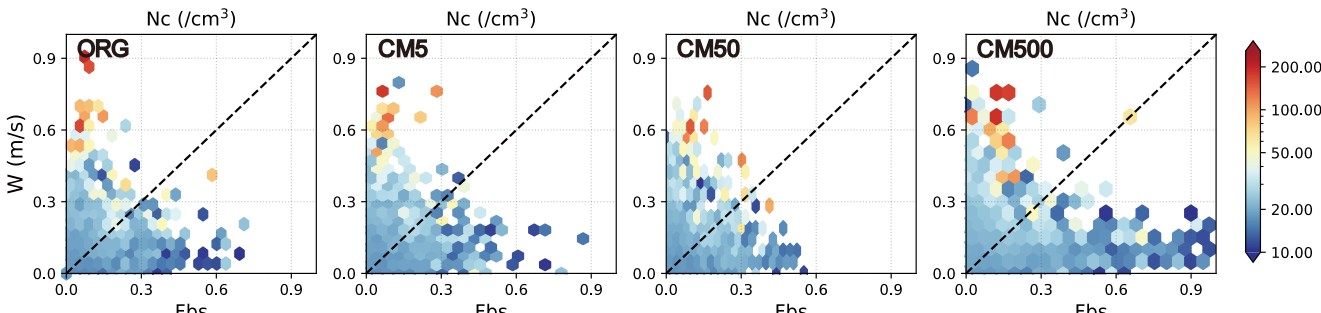

**Figure 18 Distribution of cloud droplet number concentration (Nc, /cm³) as a function of updraft strength (W, in m/s) and activation intensity (Fbs). Panels ORG, CM5, CM50 and CM500 represent simulation results under different coarse-mode aerosol concentrations, respectively. The color of the dots indicates the magnitude of Nc value. Dashed lines represent the relationship line where W/Fbs=1, indicating equal ratios of updraft strength to activation intensity.**

## 4 Discussion

The numerical simulation results of this study reflect the significant impact of the relative strength of the W and cloud droplet activation on cloud microphysical characteristics, as well as the nonlinear effects of increased coarse-mode aerosol concentration on cloud microphysical processes. In both the HRZ and LRZ intervals, the intensity of the W is negatively correlated with $\varepsilon$, consistent with findings by Korolev et al. (2016) and Chen et al. (2016, 2018), where increased supersaturation promotes cloud droplet condensational growth, leading to a convergence of Rv with increasing Fbs and a reduction in $\varepsilon$. The mechanisms of how variations in coarse-mode aerosol concentration impact cloud microphysical properties are summarized in Figure 19.

Similar to the findings of Dror et al. (2020), even at low Ncm concentrations, the cloud system demonstrates sensitivity to changes in Ncm, altering the development of cloud microphysical processes by triggering early cloud droplet collision-coalescence processes. However, different from Dror et al. (2020), the significant impact of Ncm on cloud droplet collision-coalescence process are mainly found during the vigorous development stage of warm clouds, consistent with the findings of van den Heever et al. (2006), which indicated the promotion of cloud water content by larger CCN particles during the later

stages of cloud evolution. However, the warm clouds Dror et al. (2020) studied are in marine area, while the warm cloud

process simulated in this study is a continental stratiformed warm cloud process.

With increasing Ncm concentration, the increase in Rv is accompanied by enhanced coalescence intensity. In contrast,

through satellite and ground observations in Taiwan, Chen et al. (2021) found that increased aerosol concentration leads to a

decrease in effective radius of cloud droplet, redistributing more cloud water into smaller droplets and inhibiting cloud droplet

collision-coalescence growth, possibly due to differences in aerosol spectrum involved in cloud processes.

Moreover, it can be found that the impact of increased coarse-mode aerosol concentration on cloud droplet number

concentration is nonlinear. As Ncm concentration increases, number concentration exhibits a pattern of initially decreasing

then increasing trend. Concurrently, the correlation between ε and Nc also shows nonlinear features. This nonlinear variation

may be attributed to the differences in the relative strength of the supply of small cloud droplets from aerosol activation and

the consumption of small cloud droplets through collision-coalescence processes. This contrasts with the conclusions of Lu et

al. (2006), whose numerical simulation results indicated a monotonic decrease in ε with an increasing number of concentrations

influenced by aerosol concentration.



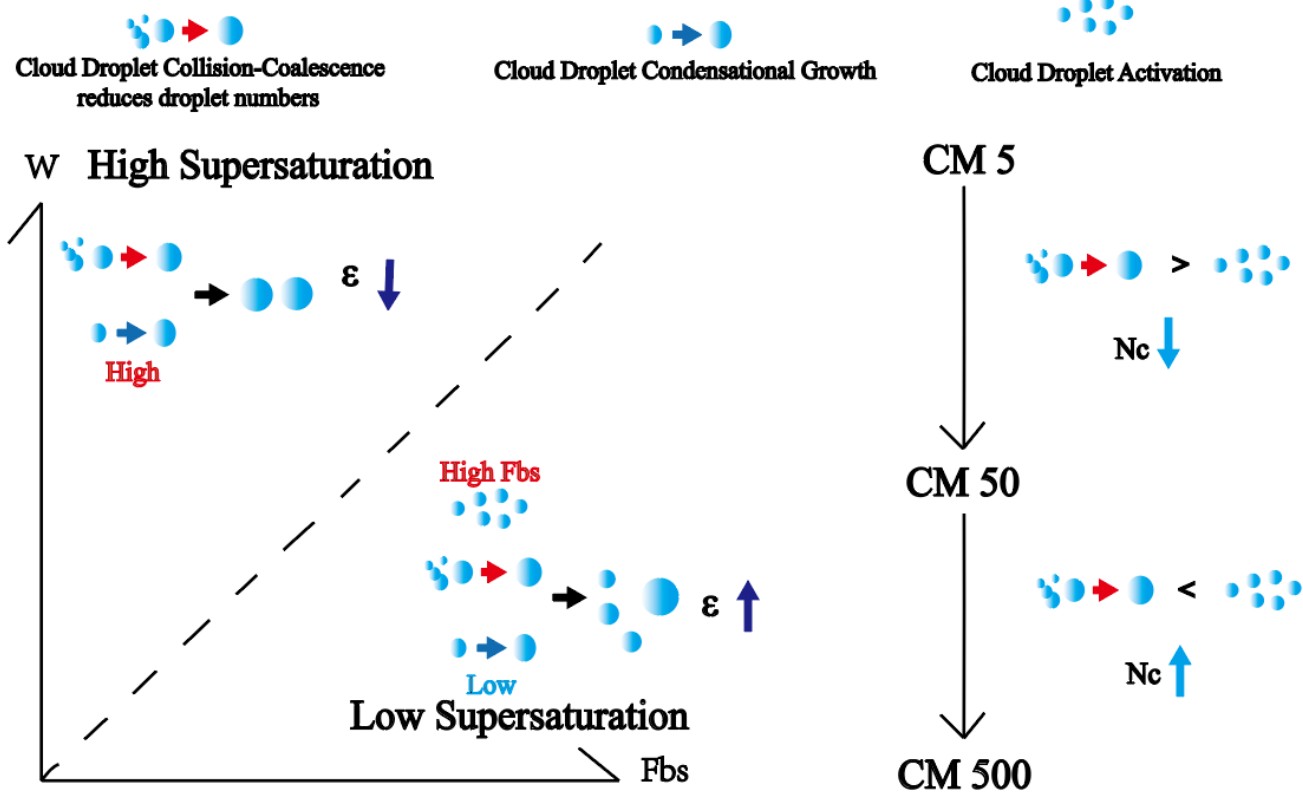

**Figure 19 Mechanisms of coarse-mode aerosol concentration impacts on microphysical processes.**

## 5 Conclusions

This study utilized the SBM-FAST scheme within the WRF model to simulate a warm cloud process in Jiangxi, China. Sensitivity experiments were conducted to analyse the influence of coarse-mode aerosol concentration on the macroscopic and microphysical characteristics of warm clouds. Additionally, the study investigated how variations in coarse-mode aerosol concentration under different W and activation intensity conditions affect ε and related microphysical processes. The specific conclusions are as follows:

(1) The cloud microphysical properties exhibit differences at various heights and time stages. During the early development stage, the CM5 experiment has a minimal impact on Nc, while in the CM50 and CM500 experiments, Nc

increases in the cloud base region. In the CM500 experiment, the increase in Nc is most significant, accompanied by a decrease in Rm in the cloud base region, indicating that more small cloud droplets are activated in the CM500 experiment. During the

vigorous development stage, Nc and Clw significantly increase in the cloud top region in the CM50 and CM500 experiments. In the CM500 experiment, both Nc and Clw are higher than in the control experiment. Additionally, the increased Ncm concentration leads to an earlier development of cloud tops above 3.5 km without synchronous growth in the maximum thickness of clouds.

(2) Increasing Ncm concentration leads to an earlier onset of collision-coalescence process and affects ε characteristics

under different Ncm concentrations. Generally, ε and T exhibit a fluctuating pattern along the vertical direction. With increasing Ncm concentration, the intense cloud droplet collision-coalescence process (T > 0.5) occurs earlier in CM500. During the vigorous development stage of the cloud, especially in the upper cloud region, T significantly increases with the increasing of Ncm, accompanied by an increase in ε, which indicates that under higher Ncm concentrations, the collision-coalescence processes and formation of large size cloud droplets are more active.

(3) The response of cloud microphysical processes to changes in Ncm is largely influenced by the relative strength of the W and cloud droplet activation, exhibiting nonlinear characteristics. In the LRZ interval, under lower supersaturation conditions, the enhancement of activation of small cloud droplets leads to a decrease in the average cloud droplet size and an increase in ε. As the Ncm concentration increases from 5 times to 50 times, the peak value of Nc decreases, but as Ncm further increases to 500 times, the peak value of Nc increases instead. This nonlinear relationship is related to the differences

in the relative strength of the supply of small cloud droplets from aerosol activation and the consumption of small cloud droplets through coalescence processes. When Ncm increases by 5 and 50 times, the cloud droplet collision-coalescence intensity increases, resulting in the rapid consumption of small cloud droplets and a decrease in ε. However, when the Ncm

concentration increases to 500 times, more aerosol particles participate in activation, replenishing the rapid consumption of small cloud droplets by the collision-coalescence process. This leads to an increase in Nc and ε.

**6 Conflict of Interest**

The authors declare that the research was conducted in the absence of any commercial or financial relationships that could be construed as a potential conflict of interest.

**7 Acknowledgments**

Firstly, we declare that the research presented herein was conducted in the absence of any commercial or financial relationships that could be construed as a potential conflict of interest. Additionally, this work has not been submitted or published in any other journal prior to this submission.

We are immensely grateful for the financial support provided by the National Natural Science Foundation of China under Grant Nos. 42061134009 and 4197517.

Special thanks are also extended to the computational platforms that were instrumental in conducting this study. This study was supported by the National Key Scientific and Technological Infrastructure project "Earth System Numerical Simulation Facility" (EarthLab). Moreover, we acknowledge the High Performance Computing Center of Nanjing University of Information Science and Technology for their support of this work.

**8 Data Availability Statement**

The data used in this study can be accessed at the following link: https://doi.org/10.57760/sciencedb.16382. The data link includes WRF model simulation results. The simulation data used in this article are described as follows:

The dataset includes the WRF model simulation output data wrfout files and the initial field data used for the simulations. The simulation output data is in .netcdf format, containing simulation data such as temperature, cloud water content, number concentration, etc. The initial field data is in .grib format, with specific information as follows:

The simulation output data includes wrfout and sbmonly files. Information of cloud and aerosol particle bins is stored in the sbmonly files. Data such as temperature, pressure, total water content of various cloud particles, and cloud number concentration are located within the wrfout files. The data format of the wrfout files and the sbmonly files is consistent, with each file containing simulation data for a duration of one hour. The temporal resolution of the data within each file is 10 minutes. Data such as temperature, air pressure, and cloud water content are stored in the form of three-dimensional or four-dimensional arrays. For example, the temperature T[6,56,375,375] is stored as a four-dimensional array, where the dimensions represent time, vertical levels, latitude, and longitude, respectively. Specific data content and descriptions can be found within the files.

In addition, the initial fields used in the numerical simulations are based on the Fifth generation of ECMWF atmospheric reanalysis of the global climate (ERA5) hourly data on pressure levels. These data can be accessed at the following link: https://cds.climate.copernicus.eu/cdsapp#!/dataset/reanalysis-era5-pressure-levels?tab=overview. The study utilized all height variables for every 6 hours from December 24th, 2014, 18:00 to December 25th, 2014, 06:00.

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
