# Peer review of "Impact of Coarse-Mode Aerosol on Jiangxi Warm Clouds Considering Different Updraft and Activation Intensities: An SBM-FAST Approach"

_EGUsphere, 2024_

## Referee Comment (RC1)

Review of "Impact of Coarse-Mode Aerosol on Jiangxi Warm Clouds Considering Different Updraft and Activation Intensities: An SBM-FAST Approach" by Yi Li and Xiaoli Liu

**Overall comments**

This manuscript provides simulations of a stratiform cloud system under varying concentrations of coarse-mode aerosols. While there is merit to studying such a topic, I think the manuscript is poorly constructed and justified, as it is hard to understand why it is being made in the first place and how it fits the context of the literature. Most important in that regard is the sensitivity experiments increasing the concentrations of the aerosol coarse mode by 5x and even up to 500x in the extreme case. I do not think those extreme increases are realistic and there is no discussion about it in the text. If there is any evidence of such changes in the region, it should be presented and discussed. There are also numerous English corrections to be made, as I point out a few of them in the "Specific comments" below. Most concerning, however, is the similarity between this manuscript and another one from mostly the same authors, that is referenced in Line 125.

After I saw the reference to the other manuscript, I started comparing them and decided to stop the detailed review I was developing because I cannot recommend the publication of a manuscript with such large overlaps with another one. I will leave the unfinished review I wrote hoping that it helps the authors, but I will not continue further.

Here is a list of similarities between the two manuscripts to justify my "reject" recommendation:

Section 2:

Since the simulation setup is basically the same between the two manuscripts, aside from the sensitivity simulations, Section 2.1 is almost identical. Figure 1 is the same in both manuscripts.

Section 2.2 in both manuscripts describe the same microphysical scheme, although with different wording.

Section 2.3 in both manuscripts describe the sensitivity experiments, the only change being the variables being modified.

Section 2.5 in this manuscript is basically the same as Section 2.4 in the other manuscript.

Section 3:

Section 3.1 in both manuscript describe the model validation, even using the same figure of the comparison between the model simulations and satellite observations

The next sections within Section 3 are all ordered in the same way between the manuscripts

Sections 4 and 5 in both manuscripts are written to fit the slightly different methodological approaches of the two manuscripts.

Based on the above, my recommendation is to reject this manuscript. My advice to the authors is to 1) rewrite both manuscripts to avoid overlaps and make it clear that they are companion manuscripts - i.e. adding Part 1 and Part 2 to the title, for instance. This would include a good justification as to why there should be 2 publications in the first place. 2) combine both manuscripts into a single one and resubmit. Whatever is the case, the manuscript would still need major revisions about the scientific results being presented. For instance, the current manuscript would have to do a better job at justifying the simulation approach and why use such extreme concentrations of coarse-mode aerosols.

**Major comments**
- The last paragraph of the introduction would benefit from additional explanations. It is unclear which macro and microcharacteristics the study will focus on. It would help contextualize the study if the text here was a bit more precise.
- Another point about the title/introduction: it should be made clearer that the study focuses on stratiform and not convective clouds early on. I would suggest even adding this information to the title of the manuscript. The reason is that it would make it easier to compare this study to others in the literature. Since I personally work with convective clouds, I read the whole introduction thinking this manuscript would be about convective clouds, which is not the case.

**Specific comments**
Line 20: the last sentence of the abstract is a bit confusing. Please rewrite it.
Line 29: consider rephrasing to avoid repeating "studies" and avoid using "lots of studies". A more formal sentence would be more adequate.
Line: please define $\varepsilon$
Line 39: updraft intensity
Line 67: modes
Line 68: please define W
Line 69: regulates -> regulating
Line 70: strengthen -> strengthening
Line 84: It -> They
Line 91: obviously -> obvious
Line 93: delete "Consequently,"
Line 95: without knowing what is the baseline level of Ncm, we cannot tell how many Ncm would be in the other simulations. But in principle, a 5x increase seems quite large already, so I was surprised to see the authors also tested a 500x increase. There should be a discussion about it in the text - probably not here, but in the description of the modeling setup. Then the authors do not have to mention the 5-500x increases in the introduction, leaving it more generic. But I would like to reinforce that this justification is very important for the manuscript - especially if a 500x is realistic or not.
Section 2.1: it would be better to present the simulation setup in a table, showing the grid setup and the parameterization options.

Line 120: ground surface -> surface

Line 122: control -> influence

Lines 123: basins -> valleys

Line 124: if this manuscript is a companion to another manuscript, it should be made clearer. Potentially even adding "Part I" and "Part II" to the title, for instance. I find it concerning that the two manuscripts share similar text and figures, especially in the introduction and the methodology sections. The same information should not be present in both manuscripts, they should rather be complementary and the complementary nature of the manuscript should be made clear from the start.

Line 138: area -> areas

Line 141: delete "is"

Section 2.5: the definition of the variables should be given here not in the supplement, given that they are central to the manuscript.

Line 188: please define $C_{lw}$. I should also note that the more common variable name is LWC (liquid water content).